# Simultaneous Improvement of Dissolution Behavior and Oral Bioavailability of Antifungal Miconazole via Cocrystal and Salt Formation

**DOI:** 10.3390/pharmaceutics14051107

**Published:** 2022-05-22

**Authors:** Ksenia V. Drozd, Alex N. Manin, Denis E. Boycov, German L. Perlovich

**Affiliations:** G.A. Krestov Institute of Solution Chemistry of the Russian Academy of Sciences, 1 Akademicheskaya St., 153045 Ivanovo, Russia; ksdrozd@yandex.ru (K.V.D.); anm@isc-ras.ru (A.N.M.); denboycov11@gmail.com (D.E.B.)

**Keywords:** antifungal, miconazole, dicarboxylic acids, grinding, slurry, lyophilization, dissolution, bioavailability

## Abstract

Miconazole shows low oral bioavailability in humans due to poor aqueous solubility, although it has demonstrated various pharmacological activities such as antifungal, anti-tubercular and anti-tumor effects. Cocrystal/salt formation is one of the effective methods for solving this problem. In this study, different methods (liquid-assisted grinding, slurrying and lyophilization) were used to investigate their impact on the formation of the miconazole multicomponent crystals with succinic, maleic and dl-tartaric acids. The solid state of the prepared powder was characterized by differential scanning calorimetry, powder X-ray diffraction and scanning electron microscopy. It was found that lyophilization not only promotes partial amorphization of both salts but also allows obtaining a new polymorph of the miconazole salt with dl-tartaric acid. The lyophilized salts compared with the same samples prepared by two other methods showed better dissolution rates but low stability during the studies due to rapid recrystallization. Overall, it was determined that the preparation method of multicomponent crystals affects the solid-state characteristics and miconazole physicochemical properties significantly. The in vivo studies revealed that the miconazole multicomponent crystals indicated the higher peak blood concentration and area under the curve from 0 to 32 h values 2.4-, 2.9- and 4.6-fold higher than the pure drug. Therefore, this study demonstrated that multicomponent crystals are promising formulations for enhancing the oral bioavailability of poorly soluble compounds.

## 1. Introduction

Miconazole (MCL, Figure 1) is a first-generation synthetic imidazole that displays a broad spectrum of antifungal activity against many Candida species [1,2]. MCL is also known to be active against Mycobacterium tuberculosis [3], and exhibits anti-tumor effects in the treatment of breast cancer [4]. Miconazole is a weak base with an extremely low solubility in water (less than 1μg·mL^−1^). Along with poor oral absorption and rapid clearance [5], it is one of the main reasons for its poor systemic activity against fungal infections [6]. Thus, the oral bioavailability of MCL in humans is 25–30% [7] compared with 55% for itraconazole [8] or 90% for fluconazole [9]. Hence, the MCL-based preparations are currently used only topically and are available in various topical formulations, including cream, lotion, spray liquid or suppository [10]. The active pharmaceutical ingredient (API) used in these preparations is applied as a nitrate salt [11]. Although the aqueous solubility of the MCL nitrate salt is much better than that of its base [12], its gastrointestinal side effects prevent the use of the salt form as solid oral dosage form [13,14]. However, even with topical use of MCL-based preparations, side effects may often occur on the application site, such as burning, redness, and swelling [15]. The use of nitrate as a counterion is one of the reasons for the presence of these side effects [13]. One possible solution to this problem is to obtain new MCL pharmaceutical salts with counterions, which can be used without limitations and not cause side effects, including from the gastrointestinal tract. This would make it possible, in the future, to use MCL preparations orally.

Previously, in an attempt to simplify counterion choice for the preparation of pharmaceutical salts, counterions suitable for oral or parenteral delivery have been divided into three classes in accordance with their GRAS (Generally Recognized as Safe) status and Accepted Daily Intake values [16,17]. According to this classification, the most preferred for the preparation of pharmaceutical salts counterions, belonging to the first class, contain physiologically ubiquitous ions and/or ions that occur as metabolites in biochemical pathways [17]. Hydrochloride and sodium are the most common counterions that have been used to prepare pharmaceutical salts in the past 20 years [18]. However, the study of the impact of different salt formers is necessary to select an optimal salt form, since the influence of a given counterion on the API’s physicochemical properties and pharmacodynamics is still difficult to predict without appropriate experiments. In addition to the above salt formers, the first class also includes some carboxylic acids, in particular aliphatic dicarboxylic acids. Previously, we have already screened MCL with a number of dicarboxylic acids, as a result of which the formation of a new two-component solid phases (salts or cocrystals) was confirmed for seven out of ten acids [19]. At the same time, according to the analysis of the Cambridge Structural Database, crystal structures have not been deciphered for all of them due to the difficulty of obtaining single crystals, as in the case of the itraconazole multicomponent crystals [20,21]. However, this is no reason not to carry on further studies of new multicomponent crystals, including the study of bioavailability in comparison with the original API, if new forms are promising [22]. From this list, three multicomponent solid forms of MCL can be selected based on the frequency of use of dicarboxylic acids in pharmaceutical formulations: cocrystal with succinic acid (SucAc) and two salts with maleic (MlcAc) or dl-tartaric (TartAc) acids (Figure 1) [17], which can become a modern alternative to the commercial form of this API. In this work, we were able to show the influence of not only a specific dicarboxylic acid on the API dissolution process but also methods used for preparing the MCL multicomponent crystals.

A great number of techniques are known today for obtaining multicomponent crystals [23]. However, researchers rarely investigate the effect of different preparation methods of multicomponent crystals on the API’s physicochemical properties and pharmacokinetics [24,25]. Researchers typically use one of the most common methods for preparing multicomponent crystals, such as grinding, while the preparation method has a great influence on the characteristics of the solid state of a chemical (crystallinity, porosity, particle size, surface area) and its physicochemical properties, respectively [23]. In this work, the MCL multicomponent solid forms were obtained via multiple methods (liquid-assisted grinding (LAG), slurrying and lyophilization) and characterized comprehensively. Lyophilization, unlike the first two methods, is used extremely rarely for the preparation of multicomponent crystals [26,27,28,29,30], considering that this method makes it possible to obtain a fine powders. In addition, the in vivo pharmacokinetic study of the three MCL multicomponent crystals was conducted for the first time in rabbits in order to evaluate their pharmaceutical applicability, which will lay a solid foundation for the further development of new dosage forms of miconazole.

## 2. Materials and Methods

### 2.1. Materials

The miconazole base was purchased from abcr GmbH. The dicarboxylic acids (succinic acid, maleic acid and dl-tartaric acid) were procured from Merck (Kenilworth, NJ, USA), Sigma-Aldrich (St. Louis, MO, USA) or Acros Organics (Geel, Belgium), respectively, and were used as received. All solvents used (tert-butanol, methanol, acetonitrile) were of analytical or chromatographic grade.

### 2.2. Sample Preparation

#### 2.2.1. Liquid-Assisted Grinding (LAG)

Physical mixtures (50–60 mg) of MCL with a dicarboxylic acid in a 2:1 (for the [MCL + SucAc] cocrystal) or 1:1 (for the [MCL + MlcAc] and [MCL + TartAc] salts) molar ratios were added to a 12 mL agate grinding jars with 10 agate balls. The mixtures were ground along with 50 μL methanol for 30 min at a rate of 500 rpm using a Fritsch planetary micro mill (Pulverisette 7).

#### 2.2.2. Slurry Experiments

Physical mixtures (120–140 mg) of MCL and dicarboxylic acid in a 2:1 (for the MCL + SucAc] cocrystal) or 1:1 (for the [MCL + MlcAc] and [MCL + TartAc] salts) molar ratios were slurred in methanol for 12 h under ambient conditions. The resulting samples were filtered and dried for 4 h.

#### 2.2.3. Lyophilization

Firstly, the physical mixtures of MCL and dicarboxylic acid (10–12 mg) were dissolved on stirring in 10 mL of aqueous solution, comprising 30 to 90% (*w*/*v*) of tert-butanol (TBA). The resulting solution was kept in a freezer at −40 °C for 10–12 h. Then, the flasks containing the frozen MCL solution with dicarboxylic acid were quickly transferred into a precooled chamber (−50 °C) of the INEY LS-500/80 freeze-dryer (Prointeh-bio, Pushchino, Russia). Primary drying was performed at a temperature −50 °C and a pressure *p* < 50 mTorr for 24 h. After that, the temperature in a chamber was increased up to 30 °C and held at this level for the next 4–6 h. All samples after the experiment were characterized by PXRD and DSC.

### 2.3. Powder X-ray Diffraction (PXRD)

PXRD analysis of the MCL multicomponent crystals was performed on a D2 Phaser diffractometer (Bruker AXS, Karlsruhe, Germany) with Cu-Kα radiation at 30 kV and 10 mA, equipped with a Lynxeye XE-T high-resolution position sensitive detector. The PXRD patterns were recorded over the range of 5–30° (2*θ*) with a step size of 0.02° and a dwell time of 1 s.

### 2.4. Differential Scanning Calorimetry (DSC)

DSC thermograms of the MCL multicomponent crystals were recorded using a DSC 4000 (Perkin Elmer, Waltham, MA, USA). Samples tested were placed in an aluminum crucible and heated from 20 °C to 250 °C at a constant rate of 10 °C·min^−1^ under a nitrogen flow of 20 mL·min^−1^.

### 2.5. Scanning Electron Microscopy (SEM)

The particle sizes and surface morphology of the powder samples were investigated by a Quattro S scanning electron microscope (Thermo Fisher Scientific, Černovice, Czech Republic). The analysis was carried out using the secondary electron mode, line-average scanning with a work distance of 10.4 mm and acceleration voltage of 5.00 kV.

### 2.6. Powder Dissolution Experiments

During the powder dissolution experiments, an excess amount of solid sample (100 mg of MCL or an MCL-equivalent amount of cocrystal or salts) was suspended in 15 mL of a pH 6.8 phosphate buffer as dissolution medium at 37.0 ± 0.1 °C. An aliquot of the suspension was withdrawn at predetermined time points (5, 10, 15, 20, 30, 45, 60, 90, 120, 180, 240, 360 min) and filtered through a 0.2 μm syringe filter (Rotilabo^®^ PTEF). The filtrate was diluted, and the MCL concentration was determined by HPLC. All dissolution experiments were performed in triplicate. After the powder dissolution experiments, the undissolved solids were filtered, dried and analyzed by PXRD.

### 2.7. In Vivo Pharmacokinetic Study

The in vivo study was conducted with prior permission from the Ministry of Health of the Russian Federation (order no. 267 from 19 June 2003). The experimental protocol followed the ethical Guidelines of the animal research ethics and Rules of laboratory practice in conducting preclinical research in the Russian Federation (GOST 51000.3-96 and 51000.4-96). Albino rabbits with a body weight 2 ± 0.3 kg were fasted for 24 h before commencing experiments. All animals had free access to water throughout the experimental period. Powder samples of MCL and its multicomponent crystals were administrated to rabbits using a gavage vehicle at a single dose corresponding 50 mg·kg^−1^ of MCL. Blood samples (about 0.5 mL) were collected from the marginal ear vein at 0.5, 0.75, 1, 1.5, 2, 3, 4, 8, 12, 24 and 32 h after oral administration. Normal heparin was used as an anticoagulant. The blood was centrifuged at 1600 rpm for 15 min and saved at −70 °C until analysis. The MCL concentration in plasma was determined by HPLC. Pharmacokinetic parameters were determined by PKSolver [31] software based on a noncompartmental model. All results were expressed as mean ± SD.

### 2.8. High-Performance Liquid Chromatography (HPLC)

The MCL concentrations were determined by an LC-20AD Shimadzu Prominence equipped with a PDA detector and a Luna C18 column with a 5 μm particle size, 150 mm length and 4.6 mm inner diameter. The mobile phase was a mixture of acetonitrile and a 0.1% aqueous solution of trifluoroacetic acid (40:60, *v/v*) with a flow rate of 1.0 mL·min^−1^ at 40 °C. The PDA detector was set at 223 nm.

## 3. Results and Discussion

### 3.1. Solid State Analysis

It is known that MCL forms seven multicomponent crystals (cocrystals and salts) with a number of aliphatic dicarboxylic [11,19]. In this work, one cocrystal and two salts of MCL with C4-dicarboxylic acids were selected from the list of previously known multicomponent crystals based on the GRAS status of coformers: [MCL + SucAc] cocrystal (2:1), [MCL + MlcAc] salt (1:1) and [MCL + TartAc] salt (1:1). These MCL multicomponent crystals were prepared via multiple methods: liquid-assisted grinding, slurry and lyophilization. In each case, all MCL multicomponent forms, regardless of the preparation method, were characterized by DSC and PXRD methods. In contrast to the first two methods for the preparation of the MCL multicomponent solid forms, the conditions were described previously [19] and reproduced in this work without any difficulties (Appendix A), the lyophilization experiments for these systems were carried out for the first time.

Lyophilization (or freeze-drying) is a unique method of processing drug compounds to obtain high-porous powders with a low density and much better solubility [32,33]. Due to the extremely low solubility of MCL base in water, binary mixtures of cosolvent (tert-butanol) and water were used for lyophilization of its multicomponent crystals. TBA is one of the most frequently used cosolvents for freeze-drying due to a number benefits, including high freezing temperature, very short sublimation time, low sublimation enthalpy, high equilibrium vapor/solid pressure values, and low toxicity [34]. To select the optimal conditions for obtaining the MCL multicomponent forms by lyophilization method, organic solvent-water (TBA/H_2_O) mixtures with different contents of TBA (30%, 60% and 90% (*w*/*v*)) were tested during the trial experiments for each system. Freeze-drying of pure MCL was also carried out, in order to analyze the effect of the cocrystal and salts obtained by this method on the API’s physicochemical properties. In contrast to multicomponent crystals, MCL is completely passed into a liquid state during lyophilization, namely a melt. It is related to the fact that the secondary drying of all the samples was carried out at 30 °C, which is almost 28 °C above the glass transition temperature of amorphous MCL (Appendix A). Thus, it is impossible to obtain the lyophilized MCL under these experimental conditions.

Photos of the obtained powders of the MCL cocrystal and salts as a result of freeze-drying are presented in Figure 2. The photographs clearly demonstrate the influence of both the preparation method and the composition of the organic solvent-water mixture on the appearance of lyophilized substances. Moreover, the mass of the samples obtained, regardless of the composition of TBA/H_2_O mixture, was the same in all cases. Based on the appearance of the lyophilized [MCL + SucAc] cocrystal (2:1), it can be assumed that sample amorphization does not occur as a result of lyophilization in spite of the composition of the TBA/H_2_O mixture. At the same time, it can be claimed that the cocrystallization of MCL with SucAc during the freeze-drying process leads to the stabilization of the system compared to the pure API. For the [MCL + MlcAc] salt (1:1), we observed a significant increase in powder volume due to a decrease in its density with increasing TBA content in the TBA/H_2_O mixture. For the [MCL + TartAc] salt (1:1), lyophilization contributed to the production of bulk powders identical to each other in spite of TBA content. In this regard, all the powders obtained as a result of lyophilization were further analyzed using the PXRD method. The PXRD patterns of the lyophilized powders for each MCL multicomponent system are shown in Appendix A. For the [MCL + SucAc] cocrystal (2:1), the PXRD patterns, regardless of the composition of the TBA/H_2_O mixture, were identical to each other, which fully corresponded to the calculated single-crystal diffraction pattern of the cocrystal without peaks corresponding to the starting components. The crystallinity of the lyophilized cocrystal was equal to 90% (Appendix A). The PXRD patterns for the MCL lyophilized salts, depending on the TBA/H_2_O mixture composition, were significantly different. In both cases, we observed both a spreading of the peaks and a reduction in their intensities, which, in turn, indicated a decrease in the degree of crystallinity of the studied samples. Thus, the lowest value of the crystallinity degree for the [MCL + MlcAc] powder (79.4%) was observed when using the maximum amount of TBA, which is consistent with the change in the density of lyophilized powders for this salt (Figure 2). The lowest crystallinity degree for the [MCL + TartAc] salt (46.0%) was achieved using 60% (*w*/*v*) of TBA in the TBA/H_2_O mixture. Moreover, based on the comparison of the PXRD patterns of the lyophilized [MCL + TartAc] salt (1:1) with that of the crystalline sample, it was found that the peaks did not correspond to the peaks of either the original components or the crystalline salt (Appendix A). Thus, it can be assumed that a new polymorphic form of the [MCL + TartAc] salt (1:1) was obtained by the lyophilization method.

Based on the data obtained, to produce the required mass of the MCL cocrystal or salts by lyophilization, the compositions of the TBA/H_2_O mixtures were used, at which the crystallinity degree of the powders was the least. We expect that it will contribute to the higher solubility of the investigated MCL multicomponent forms. As for the [MCL + SucAc] cocrystal (2:1), we used an aqueous solution comprising 60% (*w/v*) of TBA, because the experimental conditions we used did not affect the crystallinity degree of the lyophilized powders.

According to the previously published data, melting points of the MCL multicomponent crystals studied in this work are known: 119.7 ± 0.2 °C for the [MCL + SucAc] cocrystal (2:1), 140.3 ± 0.2 °C for the [MCL + MlcAc] salt (1:1) and 171.4 ± 0.2 °C for the [MCL + TartAc] salt (1:1) [19]. The DSC curves for the MCL cocrystal and salts obtained by the various methods are shown on Figure 3. Appendix A provides a comparison of the melting points and the melting enthalpy values of the MCL multicomponent crystals with the literature data. For the [MCL + SucAc] cocrystal (2:1), regardless of the preparation method, we observed one sharp endothermic peak on the DSC curve at a temperature corresponding to the melting temperature of the cocrystal. This is consistent with the identity of the PXRD patterns of the MCL cocrystal prepared via different methods. Moreover, the absence of additional endo/exo effects on the DSC curves additionally confirms the high purity of the samples under investigation. As for the [MCL + MlcAc] salt (1:1) prepared by the various techniques, we also observed similar DSC curves, which is consistent with previously published data [11,19]. However, the melting point of the salt samples prepared by LAG and lyophilization were almost 5 ± 0.2 °C lower than for the sample prepared by the slurry method. The DSC curves of the [MCL + TartAc] salt prepared by LAG and slurrying show one sharp endothermic peak, corresponding to the melting of the powder samples. The melting temperature of the lyophilized [MCL + TartAc] salt was 15.6 ± 0.2 °C lower than the melting temperature of the salt samples prepared by the other two methods. Moreover, the melting enthalpy of the lyophilized salt was almost 2-times less than the melting enthalpy of salt samples prepared by LAG or slurry methods. Additional peaks corresponding to the melting of individual compounds or desolvation were not found on the DSC curves. Such a difference in the melting temperatures of the [MCL + TartAc] salt (1:1) produced by the various methods, coupled with the presence of new peaks in the PXRD pattern for the lyophilized sample, additionally confirms our assumption about obtaining a new polymorph of the MCL salt via freeze-drying. As a result, based on the obtained data (PXRD and DSC), it can be argued that any of the methods used by us for the preparation of the MCL multicomponent crystals makes it possible to obtain high-purity powder samples, differing in the degree of crystallinity.

### 3.2. Surface Morphology

Morphology and particle size have a significant impact on the API’s physicochemical properties, including solubility and dissolution rate [35,36]. SEM analysis of the MCL multicomponent crystals produced by the different methods was carried out to study the influence of the preparation method on the morphology of the powders obtained. As a result of comparing the acquired SEM images for each MCL cocrystal/salt produced by LAG, slurrying and lyophilization, the morphological differences can be seen (Figure 4). The powder samples prepared by LAG had a much broader particle size distribution compared to the other two preparation methods. Particles of the [MCL + SucAc] cocrystal (2:1) and [MCL + MlcAc] salt (1:1) have an irregular shape, the size of which varies on average in the range from 50 to 400 μm. The rough surface of the particles obtained is the result of mechanochemical processing. In contrast to the other two multicomponent forms, the [MCL + TartAc] salt (1:1) obtained by LAG showed pronounced agglomeration of needle-shaped crystals. The average size of the agglomerates exceeded 400 μm. In contrast to the first preparation method of the MCL multicomponent crystals, the slurry method makes it possible to obtain sufficiently homogeneous phases with constant particle morphology. Moreover, the particles have a much smoother surface and smaller size: 5–10 μm for the [MCL + SucAc] cocrystal (2:1) and the [MCL + MlcAc] salt (1:1), and 2–5 μm for the [MCL + TartAc] salt (1:1). The shape of the [MCL + TartAc] salt (1:1) particles is strikingly different from the shape of the other two multicomponent systems, which, in turn, can also affect the difference in the dissolution parameters of the powders. The SEM image of the [MCL + SucAc] cocrystal (2:1) prepared by lyophilization confirms that the powder has a high degree of crystallinity. The cocrystal crystallized as large aggregates with many crystalline features, including a rough surface and irregular edges. The diameter of the aggregates varied over a wide range, and in some cases exceeded 500 μm. The morphology of the lyophilized MCL salts differed significantly compared with the cocrystal. The powders of the lyophilized MCL salts are porous agglomerates formed by perforated flat particles, the size of which did not exceed 1 μm. Thus, it can be assumed that the lyophilized MCL salts will show the greatest increase in API solubility based on the data obtained as a result of the morphology and particle size analysis.

### 3.3. Physical Stability

Considering that the lyophilized MCL salts were partially amorphous (Appendix A), it was necessary to evaluate their relative stability during storage under ambient conditions. The physical stability of the lyophilized MCL salts was examined by PXRD analysis. The crystallization of the samples stored at room temperature was monitored at regular intervals. The PXRD patterns of the [MCL + MlcAc] and [MCL + TartAc] salts produced by lyophilization before and after storage in comparison with the PXRD patterns corresponding to their crystalline forms are presented in Figure 5.

It should be noticed that the traces of recrystallization for all lyophilized MCL salts began to appear after one day of storage, as evidenced by the appearance of some diffraction peaks (Figure 5). The intensity of the diffraction peaks increased over time, which directly affected the increase in the crystallinity degree of the analyzed samples. At the same time, the peaks for the lyophilized [MCL + TartAc] salt on the PXRD patterns do not correspond to the peaks of the crystalline salt prepared by LAG or slurry methods. Thus, the new polymorphic form of the [MCL + TatAc] salt (1:1) is stable during recrystallization. The crystallinity degree of the lyophilized salts after 1 week of storage increased from 79.4% to 86.1% for [MCL + MlcAc] and from 46.0% to 62.7% for [MCL + TartAc].

The results obtained showed that the physical stability of both lyophilized salts is extremely low, and they can easy recrystallize during storage under ambient conditions. Moreover, it can be assumed that the lyophilized MCL salts will also be the least stable during dissolution in an aqueous solution compared to their crystalline samples prepared by LAG or slurrying. Therefore, the in vitro powder dissolution experiments with the lyophilized MCL salts were performed only on the freshly prepared samples. In the future, in order to obtain more storage stable MCL salts, we will not exclude the use of various stabilizers; however, we did not set this goal in our work.

### 3.4. In Vitro Dissolution Studies

After oral administration, drugs must dissolve well in the intestinal fluid in order to enter the bloodstream. Poor solubility is one of the main reasons for limiting the API amount that is effectively absorbed. As is known, the MCL base is practically insoluble in water (~0.4×10^−2^ μg·mL^−1^ [19]). However, we have previously shown the effectiveness of cocrystallization and salt formation on the MCL hydrophilic nature, despite that fact that these multicomponent crystals dissociate in water over time [19]. In the present work, the in vitro dissolution of the MCL multicomponent crystals prepared by multiple methods was studied to evaluate the effect of a particular dicarboxylic acid and preparation method on the level and duration of drug supersaturation in water. The powder dissolution experiments of the MCL multicomponent crystals were made in aqueous solution at pH 6.8 and 37 °C. The time profiles of the MCL dissolved concentrations for each multicomponent form are shown in Figure 6, and the results are summarized in Table 1. The absence of a dissolution profile for the original MCL was due to the fact that the solubility of MCL in the studied medium was so low that it was not possible to determine it correctly.

According to the obtained dissolution profiles, all MCL multicomponent forms, regardless of the preparation method, are metastable in an aqueous solution, as evidenced primarily by the presence of a “spring and parachute’’ type effect in each case. This statement was also confirmed by PXRD analysis of the solid state after the experiment (Appendix A). However, if for both salts in the bottom phase there was only a crystalline MCL hemihydrate, then in the case of [MCL + SucAc] (2:1), along with the traces of the crystalline MCL hemihydrate, the traces of the initial cocrystal were also present, even after 24 h of the sample being in the buffer solution.

As expected, the preparation method of the MCL multicomponent crystals made a significant contribution to the difference in dissolution profiles for a particular system. The lyophilized salts showed a much higher initial dissolution rate compared to their crystalline forms obtained by LAG or slurry methods. The MCL maximum concentration (*C_max_*) was reached in 0.2 ± 0.1 h for the [MCL + MlcAc] (1:1) or 0.1 ± 0.05 h for [MCL + TartAc] (1:1) (*T_max_* values are presented in Table 1). The *C_max_* value for the lyophilized [MCL + TartAc] salt (1:1) was more than 40 times higher than for the [MCL + MlcAc] salt (1:1). However, the increased MCL concentration for these salts was not maintained and decreased rapidly over time. After 6 h, the MCL concentration decreased by almost 2 times for [MCL + MlcAc] and 13 times for [MCL + TartAc]. This increase in solubility and dissolution rate of MCL was facilitated precisely by the fact that the obtained powders of the MCL salts via lyophilization had a porous structure, which thereby led to an increase in the contact surface of the drug substance with the solvent. The MCL salts prepared by LAG and slurry methods had a much lower dissolution rate. *C_max_* values of MCL were achieved after 1.3 ± 0.5 or 1.4 ± 0.5 h for [MCL + MlcAc] and 3.2 ± 0.6 or 1.7 ± 0.5 h for [MCL + TartAc], respectively (Table 1). A subsequent decrease in the MCL concentration for these salts was much more gradual than for the lyophilized forms, thus providing a more comfortable window of the period required for API release [37]. In contrast to the [MCL + TartAc] salt, the dissolution profiles for the [MCL + MlcAc] salt produced by both LAG and slurry methods were almost identical, as evidenced by the *AUC*_0–6_ (area under the curve) values of 10.6 and 11.9 μg·mL^−1^·h, respectively. The values of *C_max_* and *AUC*_0–6_ for the [MCL + TartAc] salt prepared by slurrying were almost 5- and 3.5-times higher, respectively, compared to the same salt prepared by LAG. This difference in the dissolution parameters may be due precisely to significant differences in the morphology of the powders obtained using these methods (Figure 4). We believe that the crystal agglomeration, characteristic of the grinded salt, is one of the reasons preventing the MCL release from the salt form. Nevertheless, the level of the MCL supersaturation in the aqueous solution was the highest for the [MCL + TartAc] salt (1:1), regardless of its preparation method.

The influence of the different preparation methods on the [MCL + SucAc] cocrystal (2:1) dissolution was not as significant as for the two MCL salts. The highest *C_max_* and *AUC*_0–6_ values of the API are achieved for the cocrystal produced by slurry method, and the lowest values for the lyophilized sample. We assume that this may be related to the influence of the cocrystal preparation method on the particle size of the powder samples.

### 3.5. In Vivo Pharmacokinetic Study

After satisfactory in vitro dissolution studies, the in vivo pharmacokinetic investigations of the pure MCL and its cocrystal and salts were carried out for the first time to evaluate whether the in vitro dissolution advantages of the MCL multicomponent crystals can translate into in vivo oral bioavailability advantages. In vivo studies have been performed on rabbits, which are often used as an animal model for pharmacokinetic studies.

Based on the dissolution profiles, as well as the physical stability of the powders during storage and in aqueous media of the MCL multicomponent crystals prepared by the multiple methods, the in vivo studies were carried out for the samples produced by slurry method. The time profiles of the MCL plasma concentration after oral administration of the pure API or its multicomponent crystals to rabbits are shown in Figure 7. Calculated pharmacokinetic parameters, such as maximum MCL plasma concentration (*C_max_*), time to reach *C_max_* (*T_max_*) and area under the MCL plasma concentration versus time cure (*AUC*_0–32_), are summarized in Table 2.

For the pure MCL, the *C_max_* и *AUC*_0–32_ values are only 120 ± 20 ng·mL^−1^ and 1372 ± 420 ng·h·mL^−1^, indicating poor oral absorption due to its extremely low intrinsic solubility. However, the MCL cocrystal and salts showed significantly higher absorption compared to pure API, which is consistent with previously in vitro data. The *C_max_* values for [MCL + SucAc], [MCL + MlcAc] и [MCL + TartAc] were 2.2, 3- and 3.2-times higher, respectively, than the pure MCL. Due to that, the *AUC*_0–32_ values also increased several times compared to the original API, namely 2.4 times for the cocrystal, 2.9 and 4.6 times for the salts. Moreover, while the *C_max_* value of the [MCL + TartAc] salt (1:1) is only 14 ng·mL^−1^ higher that of the [MCL + MlcAc] salt (1:1), the *AUC*_0–32_ value for it is almost 40% higher. It is associated with a much more gradual decrease in the MCL concentration over time for the [MCL + TartAc] salt. *T_max_* for the [MCL + TartAc] salt was also far different and is 6 h instead of 1.5–2 h for pure MCL or two other MCL multicomponent crystals, referring to the prolonged effect of the salt. Thus, the MCL multicomponent crystals with C4-dicarboxylic acids studied in this work are promising for the development of new oral forms of MCL.

## 4. Conclusions

In this work, the influence of the preparation methods (liquid-assisted grinding, slurrying and lyophilization) on the solid state and dissolution of the miconazole multicomponent crystals with succinic, maleic and dl-tartaric acids was studied. It was found that the preparation method significantly affects both the morphology and size of the particles. It was revealed that the lyophilization of both miconazole salts leads to their partial amorphization, in contrast to the cocrystal. However, while the dissolution rate and the miconazole maximum concentration value in the aqueous solution for the lyophilized salts was many times higher than for the same samples prepared by grinding and slurrying, their physical stability was extremely low. The in vivo pharmacokinetic study of the miconazole cocrystal and salts was carried out for the first time. It was found that the improved dissolution parameters of miconazole are successfully converted into improved in vivo pharmacokinetic profiles. It justified the promise of the miconazole multicomponent forms in the development of new oral dosage forms based on it.

## Figures and Tables

**Figure 1 pharmaceutics-14-01107-f001:**
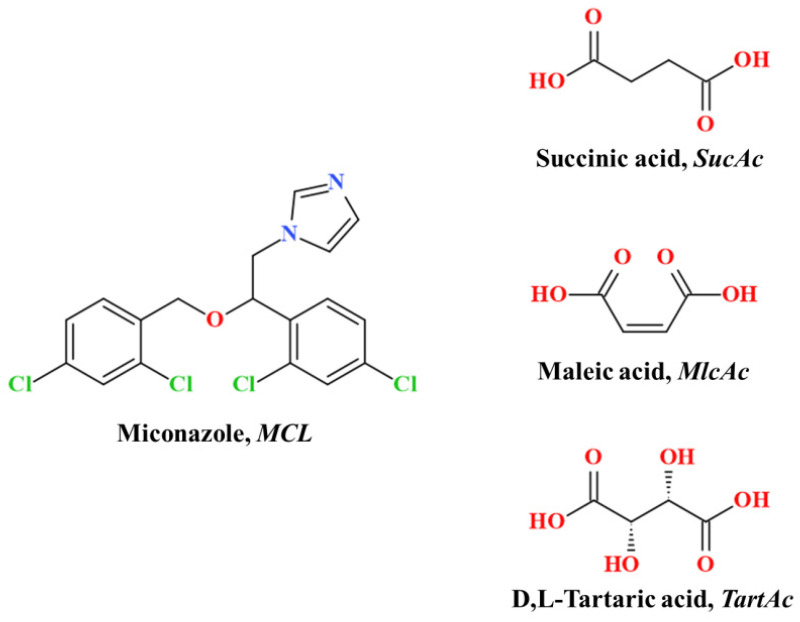
Chemical structures of miconazole and dicarboxylic acids used in this study.

**Figure 2 pharmaceutics-14-01107-f002:**
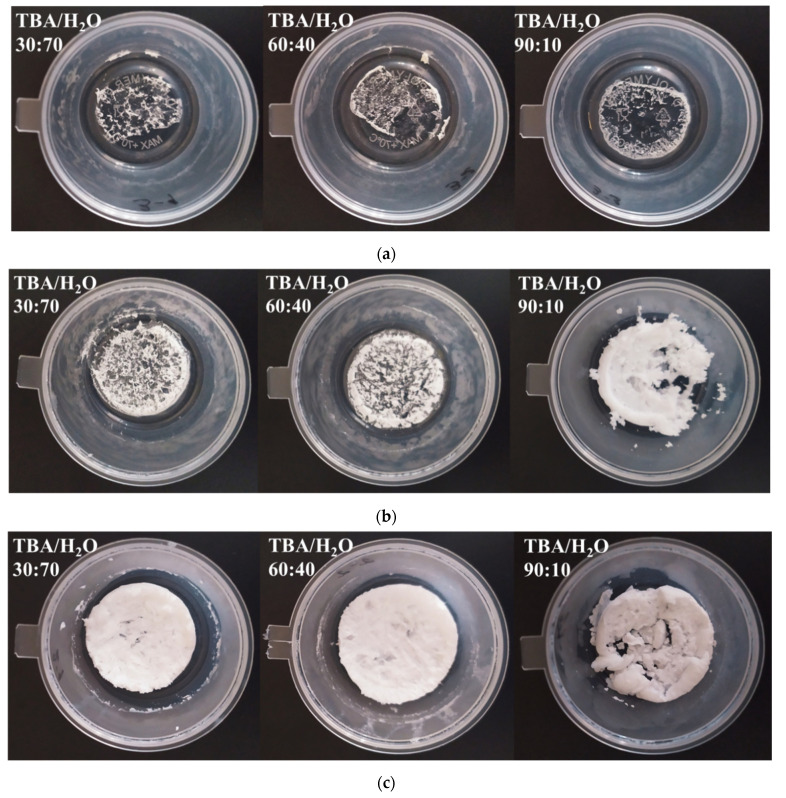
Comparison of the freeze-dried powders of (**a**) [MCL + SucAc] cocrystal (2:1), (**b**) [MCL + MlcAc] salt (1:1) and (**c**) [MCL + TartAc] salt (1:1) obtained from TBA/H_2_O mixtures with different TBA content.

**Figure 3 pharmaceutics-14-01107-f003:**
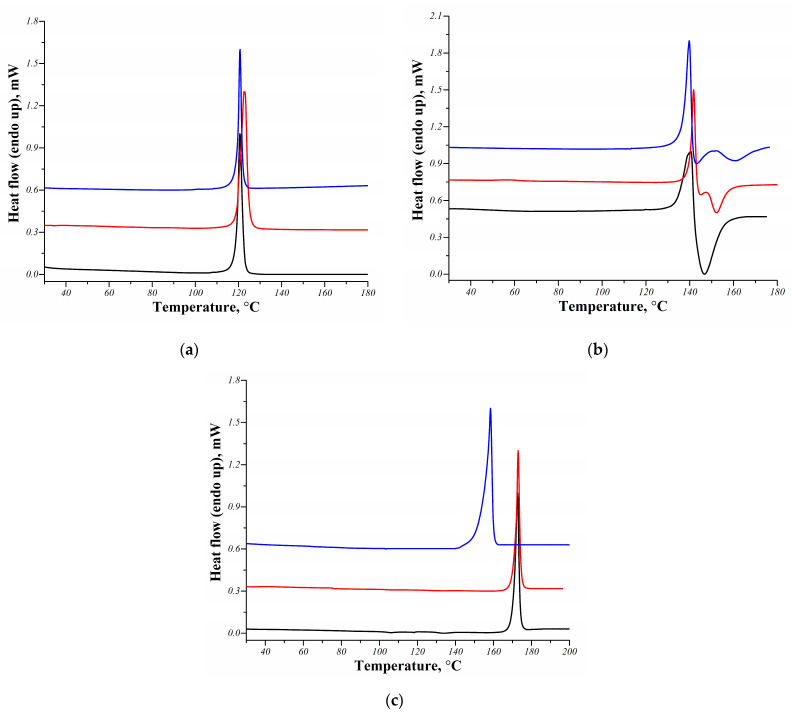
DSC curves of the MCL multicomponent crystals: (**a**) [MCL + SucAc] cocrystal (2:1), (**b**) [MCL + MlcAc] salt (1:1), (**c**) [MCL + TartAc] salt (1:1) prepared via LAG (black line), slurrying (red line) and freeze-drying (blue line).

**Figure 4 pharmaceutics-14-01107-f004:**
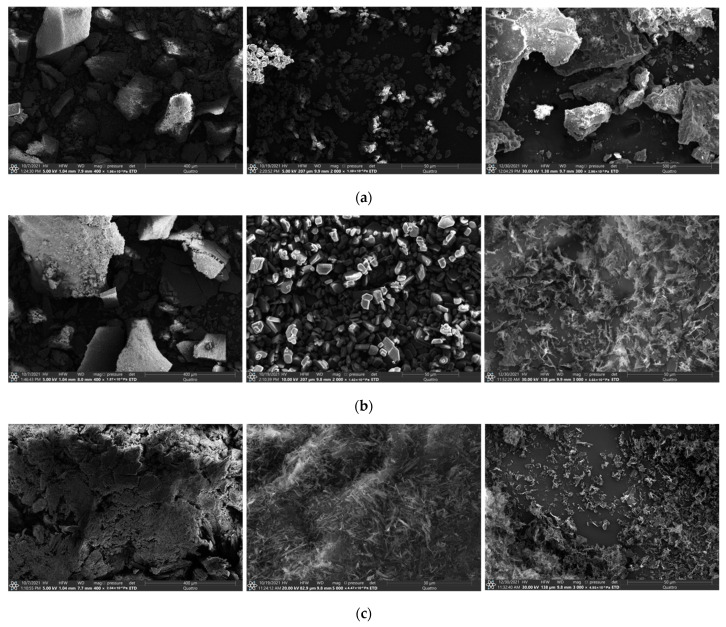
SEM images of the MCL multicomponent crystals: (**a**) [MCL + SucAc] cocrystal (2:1), (**b**) [MCL + MlcAc] salt (1:1) and (**c**) [MCL + TartAc] salt (1:1), prepared by LAG, slurring and freeze-drying (from left to right).

**Figure 5 pharmaceutics-14-01107-f005:**
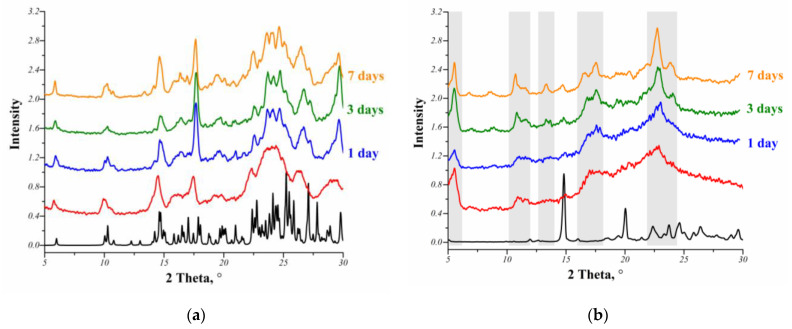
PXRD patterns for the MCL salts: (**a**) [MCL + MlcAc] (1:1) and (**b**) [MCL + TartAc] (1:1) in crystalline forms (black line) and freeze-dried samples (red line) studied as a function of storage time. Grey stripes highlight some unique peaks different from the peaks of the crystalline [MCL + TartAc] salt.

**Figure 6 pharmaceutics-14-01107-f006:**
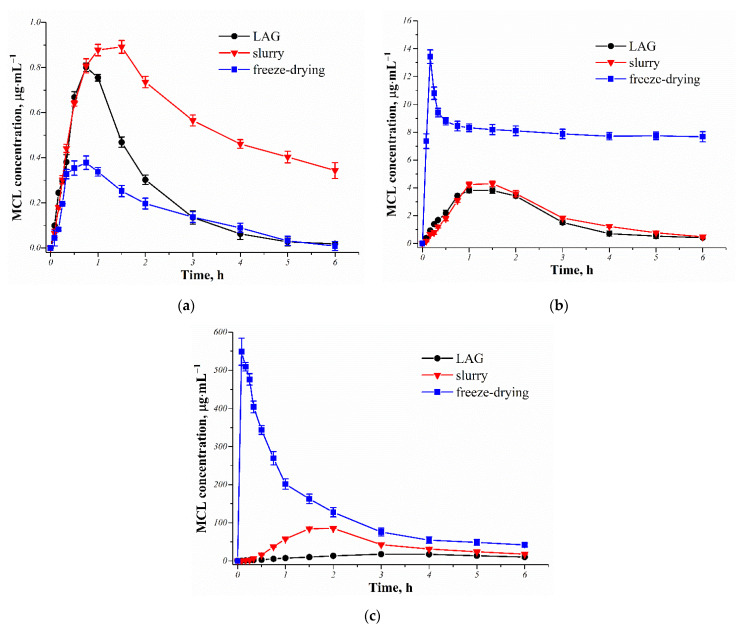
Time profiles of the MCL dissolved concentrations from its multicomponent crystals prepared via different methods in aqueous solution pH 6.8 at 37 °C: (**a**) [MCL + SucAc] (2:1), (**b**) [MCL + MlcAc] (1:1), (**c**) [MCL + TartAc] (1:1).

**Figure 7 pharmaceutics-14-01107-f007:**
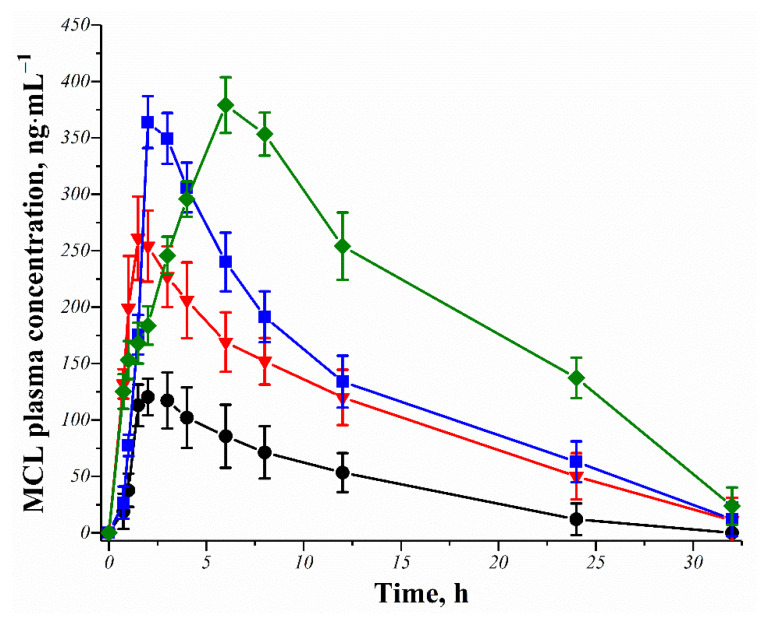
Mean plasma concentration-time profiles of the MCL and its multicomponent crystals after oral administration to rabbits. Key: –●–—pure MCL, –▼–—[MCL + SucAc] (2:1), –■–—[MCL + MlcAc] (1:1), –♦–—[MCL + TartAc] (1:1).

**Table 1 pharmaceutics-14-01107-t001:** Dissolution results of the MCL multicomponent crystals prepared by the multiple methods in aqueous solution pH 6.8 at 37 °C.

MCL Multicomponent Crystal	Preparation Method	Dissolution Profile				Residue Form at 24 h
		T_max_, h	C_max_, μg·mL^−1^	C_6h_, μg·mL^−1^	AUC_0–6_, μg·mL^−1^·h	
[MCL + SucAc] (2:1)	LAG	0.7 ± 0.2	0.80 ± 0.02	(1.78 ± 0.23) × 10^−2^	1.42	cocrystal, MCL·H_2_O
	slurry	1.2 ± 0.4	0.89 ± 0.03	0.34 ± 0.06	3.36	
	freeze-drying	0.7 ± 0.3	0.39 ± 0.03	(0.85 ± 0.03) × 10^−2^	0.90	
[MCL + MlcAc] (1:1)	LAG	1.3 ± 0.5	3.82 ± 0.17	0.41 ± 0.07	10.6	cocrystal, MCL·H_2_O
	slurry	1.4 ± 0.5	4.30 ± 0.20	0.48 ± 0.09	11.9	
	freeze-drying	0.2 ± 0.1	13.43 ± 0.38	7.67 ± 0.26	48.2	
[MCL + TartAc] (1:1)	LAG	3.2 ± 0.6	17.45 ± 0.40	10.13 ± 0.50	73.3	cocrystal, MCL·H_2_O
	slurry	1.7 ± 0.5	85.26 ± 3.05	17.71 ± 0.60	248.2	
	freeze-drying	0.1 ± 0.05	548.60 ± 24.31	41.84 ± 2.10	769.7	

**Table 2 pharmaceutics-14-01107-t002:** Main pharmacokinetic parameters of pure MCL and its multicomponent crystals in rabbits.

	MCL	[MCL + SucAc] (2:1)	[MCL + MlcAc] (1:1)	[MCL + TartAc] (1:1)
*C_max_*, ng·mL^−1^	120 ± 20	261 ± 37	365 ± 23	379 ± 30
*T_max_*, h	2 ± 0.6	1.3 ± 0.6	2.2 ± 0.5	6.2 ± 0.6
*AUC*_0–32_, ng·mL^−1^·h	1372 ± 418	3296 ± 537	4017 ± 617	6349 ± 661

## Data Availability

The results obtained for all experiments performed are shown in the manuscript and Appendix A, the raw data will be provided upon request.

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
