# Peer review of "Simultaneous Improvement of Dissolution Behavior and Oral Bioavailability of Antifungal Miconazole via Cocrystal and Salt Formation"

_pharmaceutics, 2022, doi:10.3390/pharmaceutics14051107_

Round 1

Reviewer 1 Report

On the one hand the study is rich, properly conducted and presents a lot of interesting observations. On the other hand it requires some crucial revisions. My detailed comments are listed below.

In the introduction the Authors do not mention that MCL in a form of base exists as a hemihydrate, the crystal structure of this hemihydrate is known since 1979. I assume that the Authors have used hemihydrate form of MCL as the starting material, am I correct?

Lines 33-36, this sentence is grammatically incorrect.

Line 99, why the Authors have used 2:1 ratio for the [MCL+SucAc] since the 1:1 cocrystal is a well known form  (CSD refocde EVATED)?

Line 135, did the Authors ensure that the particle sizes of the studied samples are similar? The particle size has a great impact on the dissolution behavior.

Lines 171-186, this should be the part of Introduction

Line 234, I wouldn’t call this a new polymorphic form. The obtained phase is surely amorphic, therefore it is not a new polymorphic phase as it is not crystalline. Even if the Authors assume that this is a way to obtain a new polymorphic crystalline form, this statement requires stronger proof (i.e. ssNMR, SCXRD).

Figure 3, why the Authors have not calculated the enthalpies of those phase transitions and compared them with the literature values?

Solid state analysis: I really wish the Authors had performed some more solid state analysis, i.e. TGA, FT-IR, ssNMR… This would enable more credible discussion and significantly improve the quality of the submitted manuscript.

Lines 360-362, how can you be sure that was a monohydrate?

Table 1, there are some statistical methods that can validate whether there are significant differences between the obtained profiles. Why the Authors have not used them?

Author Response

Response to Reviewer 1

On the one hand the study is rich, properly conducted and presents a lot of interesting observations. On the other hand it requires some crucial revisions. My detailed comments are listed below.

Reviewer comment #1

In the introduction the Authors do not mention that MCL in a form of base exists as a hemihydrate, the crystal structure of this hemihydrate is known since 1979. I assume that the Authors have used hemihydrate form of MCL as the starting material, am I correct?

Authors’ response

In the present work, we have used the anhydrate form of miconazole as the starting material (СAS 22916-47-8). The purity of the starting material has been confirmed by DSC/TG and PXRD analysis. Moreover, the crystal structure of the miconazole anhydrate was first reported in 2022 (doi: 10.1021/acs.cgd.2c00112). DSC/TG curves and overlay of experimental and calculated PXRD patterns for miconazole used in this study and miconazole anhydrate are presented below (Figure 1). The absence of dehydration stage during heating of the sample and the compatibility of experimental and calculated PXRD patterns proves that we used anhydrate form of miconazole.

(a)

(b)

Figure 1. (a) DSC/TG curves of miconazole used in this study and (b) overlay of experimental and calculated PXRD patterns for miconazole anhydrate.

Reviewer comment #2

Lines 33-36, this sentence is grammatically incorrect.

Authors’ response

The text has been corrected.

Reviewer comment #3

Line 99, why the Authors have used 2:1 ratio for the [MCL+SucAc] since the 1:1 cocrystal is a well known form (CSD refocde EVATED)?

Authors’ response

The [MCL+SucAc] cocrystal (2:1) is the only known form for this multicomponent crystal (CSD refcode EVATAZ). The CSD refcode EVATED corresponds to the miconazole cocrystal with fumaric acid in a 2:1 molar ratio.

Reviewer comment #4

Line 135, did the Authors ensure that the particle sizes of the studied samples are similar? The particle size has a great impact on the dissolution behavior.

Authors’ response

In the present study, both morphology and particle size of the studied miconazole cocrystal or salts prepared via different methods have been studied. In Section 3.2, we have shown that the preparation method has a direct impact on morphology and particle sizes of the studied powders. In Section 3.4, we have shown how it affects the dissolution behavior of each miconazole multicomponent crystal.

Reviewer comment #5

Lines 171-186, this should be the part of Introduction

Authors’ response

Thanks for your comment. The text has been corrected.

Reviewer comment #6

Line 234, I wouldn’t call this a new polymorphic form. The obtained phase is surely amorphic, therefore it is not a new polymorphic phase as it is not crystalline. Even if the Authors assume that this is a way to obtain a new polymorphic crystalline form, this statement requires stronger proof (i.e. ssNMR, SCXRD).

Authors’ response

In the present work, we assume that the obtained phase via lyophilization is a new polymorphic form of the miconazole salt with tartaric acid. This assumption has been confirmed by two techniques: DSC and PXRD. First of all, we have found that the melting point of lyophilized salt is different from the melting point of the miconazole salt obtained via liquid-assisted grinding or slurry. Secondly, due to the lyophilized salt is partially amorphous, on PXRD pattern we didn’t see a broad amorphous halo, but some peaks. As a result of comparing PXRD patters of the lyophilized salt and crystalline salt prepared via other methods, we found that the peaks differ. Despite all our numerous attempts, we could not obtain single crystals for any of the miconazole salt with tartaric acid forms.

Reviewer comment #7

Figure 3, why the Authors have not calculated the enthalpies of those phase transitions and compared them with the literature values?

Authors’ response

We agree with the comment. Table with thermophysical data has been added in Supporting information (Table S2).

Reviewer comment #8

Solid state analysis: I really wish the Authors had performed some more solid state analysis, i.e. TGA, FT-IR, ssNMR… This would enable more credible discussion and significantly improve the quality of the submitted manuscript.

Authors’ response

In view of the fact that all miconazole multicomponent crystals studied by us in this work are not obtained for the first time, we did not consider it necessary to provide additional studies beyond those that were presented. In the present work, we were interested in studying how different methods of preparing multicomponent crystals can affect the morphology and particle size of powders, as well as their dissolution behavior.

Reviewer comment #9

Lines 360-362, how can you be sure that was a monohydrate?

Authors’ response

We agree the solid phase corresponds to a hemihydrate. The text has been corrected.

Reviewer comment #10

Table 1, there are some statistical methods that can validate whether there are significant differences between the obtained profiles. Why the Authors have not used them?

Authors’ response

In our work, comparing of the dissolution profiles of the miconazole multicomponent crystals prepared by different methods have been via some parameters, such maximum MCL concentration (Cmax), time to reach Cmax (Tmax) and area under the MCL concentration versus time cure (AUC0-6).

Reviewer 2 Report

Give the rationale for use of rabbits for these studies.

Tmax should be expressed as median and range.

Give the components of the gavage vehicle.

Fig 6. Give the semilog plot for PK curve.

Change the x-axis to hours.

Provide the label in the figure itself and not the caption.

Table 1/2. Give the Tmax as the median and range for all treatment groups.

Justify the rationale for AUC0-6? Please give the AUClast for all the treatments.

Change the unit of the y-axis for PK curves to ng/mL.

Author Response

Response to Reviewer 2

Reviewer comment #1

Give the rationale for use of rabbits for these studies.

Authors’ response

Rabbit is one of the most often experimental models (besides other small animals like mice or rats) used for the evaluation of drugs in vivo. Moreover, in vivo pharmacokinetics of the miconazole marketed capsules have already been studied in rabbits (doi: 10.2147/IJN.S100625).

Reviewer comment #2

Tmax should be expressed as median and range.

Authors’ response

The values have been corrected.

Reviewer comment #3

Give the components of the gavage vehicle.

Authors’ response

In the present study, water was used as the vehicle to facilitate oral gavage.

Reviewer comment #4

Fig 6. Give the semilog plot for PK curve.

Change the x-axis to hours.

Provide the label in the figure itself and not the caption.

Authors’ response

Figure 6 has been corrected according Reviewer comment.

Reviewer comment #5

Table 1/2. Give the Tmax as the median and range for all treatment groups.

Authors’ response

The values have been corrected.

Reviewer comment #6

Justify the rationale for AUC0-6? Please give the AUClast for all the treatments.

Authors’ response

The parameter AUC0-6 used in Section 3.4 is defined as area under the curve from 0 to 6 h (the time of the last measured concentration). So, AUC0-6 is AUClast.

Reviewer comment #7

Change the unit of the y-axis for PK curves to ng/mL.

Authors’ response

The unit of the y-axis for PK curves has been changed according Reviewer comment.

Round 2

Reviewer 1 Report

The Authors have revised and improved their manuscript. This version can surely be accepted for publication.